# New Preservative-Free Formulation for the Enhanced Ocular Bioavailability of Prostaglandin Analogues in Glaucoma

**DOI:** 10.3390/pharmaceutics14020453

**Published:** 2022-02-20

**Authors:** Gabriel Alviset, Yohann Corvis, Karim Hammad, Josiane Lemut, Marc Maury, Nathalie Mignet, Vincent Boudy

**Affiliations:** 1Unither Développement Bordeaux, ZA Tech Espace, av. Toussaint Catros, 33185 Le Haillan, France; gabriel.alviset@unither-pharma.com; 2Faculté de Santé de Paris, CNRS, INSERM, UTCBS, 75006 Paris, France; yohann.corvis@u-paris.fr (Y.C.); nathalie.mignet@u-paris.fr (N.M.); 3Assistance Publique Hôpitaux de Paris (AP-HP), Agence Générale des Équipements et Produits de Santé (AGEPS), Département de Recherche et Développement Pharmaceutique (DRDP), 7 rue du fer à Moulin, 75005 Paris, France; 4Faculté de Santé de Paris, CNRS, CiTCoM, 75006 Paris, France; karim.hammad@u-paris.fr; 5CMC Expert, 84 rue Maurice Béjart, 34080 Montpellier, France; jlemut@cmcexpert.fr; 6Unither Pharmaceuticals, 3-5 rue St-Georges, 75009 Paris, France; marc.maury@unither-pharma.com

**Keywords:** ophthalmic drug delivery, prostaglandin analogues, micellar solubilization, bioavailability enhancement, sodium hyaluronate, polysorbate

## Abstract

Glaucoma is a wide-spread eye disease caused by elevated intraocular pressure. Uncontrolled, this pressure may lead to damages to the optic nerve. Prostaglandin analogues, such as latanoprost and travoprost (which are water-insoluble active substances), are the most used class of active pharmaceutical ingredient. To administer them as eye drops, preservatives, such as benzalkonium chloride, are used as solubilizers. The latter is known to cause a local inflammation when used chronically and is not recommended for patients with ocular surface disorders. In this work, we sought to use polysorbate 80 (PS80) as a solubilizing agent simultaneously with sodium hyaluronate (NaHA) as a thickener and cytoprotective agent for the corneal surface. The first part of this study assessed the compatibility of the excipients with the active substance, using physicochemical methods such as spectra fluorescence and differential scanning calorimetry (DSC), as well as the solubilization mechanism of PS80 regarding prostaglandin analogues using nuclear magnetic resonance (NMR). The second part evaluated the stability of a formula candidate, its viscosity upon instillation, and its pharmacokinetic profile in rabbits as compared to the commercially approved medicine Travatan^®^. The results show that sodium hyaluronate is inert with respect to travoprost, while PS80 successfully solubilizes it, meaning that benzalkonium chloride is no longer required. Moreover, the pharmacokinetic profiles of the rabbits showed that the original formula described in the present study enhanced the ocular bioavailability of the drug, making it a promising product to control intraocular pressure with a potential reduced dosage of travoprost, therefore minimizing its related side effects.

## 1. Introduction

Primary open-angle glaucoma (POAG) is the second leading cause of blindness, affecting over 50 million people worldwide [1,2]. POAG is mainly induced by the accumulation of aqueous humour in the anterior chamber of the eye, causing elevated intraocular pressure (IOP). The increase in IOP causes mechanical stress to the posterior segment of the eye, and is correlated to retinal ganglion cell death [3]. POAG is insidious, as it is painless, i.e., the patients do not feel symptoms until damage is already done to the optic nerve fibers, resulting in irreversible vision loss [4,5]. Therefore, management of IOP is essential for the wellbeing of POAG patients.

Various pharmacological agents exist, either reducing aqueous humour production (beta-blockers, alpha agonists, carbonic anhydrase inhibitors) or enhancing its elimination (prostaglandin analogues, muscarinic agonists) [6,7]. Today, the two most effective pharmacological classes are beta-blockers and prostaglandin analogues, which both present some drawbacks [7,8]. On one hand, beta-blockers are contraindicated for patients with asthma or with a history of cardiac disease. On the other hand, prostaglandin analogues rely on preservatives, such as benzalkonium chloride (BAK) or polyquaternium, for their solubilization which can cause redness of the eye as well as corneal inflammation upon repeated administration [7,9,10,11,12,13,14]. The choice of treatment in the first intention is left to the practitioner, often leading to the use of prostaglandin analogues [8,15].

With current packaging technologies, such as single-dose or preservative-free eye drops (PFED) in multidose bottles, the use of preservatives is outdated. In terms of efficacy, PFED formulations are not inferior to formulas containing BAK [13]. Therefore, prostaglandin analogues are reformulated without the use of preservatives as stabilizers and solubilizers [10,16], often using ricin oil or propylene glycol as cosolvent [12]. Indeed, PFED formulations were shown to have a better ocular tolerance, especially regarding surface parameters such as an increased tear break-up time or less burning/stinging sensation upon instillation [11,12,13]. Better ocular tolerance is particularly important as it will lead to a better patient adherence and a more successful management of the IOP, hence delaying onset of irreversible symptoms such as vision loss. Polysorbate 80 (PS80) is a surfactant used in the commercial eye-drops Restasis^®^ as a solubilizer for cyclosporine [17,18], it could be used as a substitute for BAK and potentially be better tolerated than both BAK and ricin oil, as brought to light by Fukuda et al. during in vitro/in vivo studies [19,20].

Another key excipient identified for the development of a prostaglandin analogue formulation is sodium hyaluronate (NaHA). Indeed, this high molecular weight polymer has two interesting features. First, it can increase the viscosity of the final formulation which is considered an asset for ocular drug delivery, as it can compensate for the fast clearance of the tear film [21]. Secondly, NaHA has been proven to have a cytoprotective effect on the corneal surface [22,23,24], which can be altered by prostaglandin analogue treatment [25]. Therefore, the aim of this work is to evaluate PS80 as a solubilizing agent for prostaglandin analogues, coupled with NaHA for its cytoprotective and thickening effects.

## 2. Materials and Methods

### 2.1. Materials

Pyrene 98% purity was purchased from Acros Organics (Illkirch, France). Latanoprost and travoprost were kindly provided by Sanofi Chinoin (Budapest, Hungary). High molecular weight sodium hyaluronate (NaHA) SZE grade, with an intrinsic viscosity of 2.5 m^3^/kg was purchased from Shiseido CO., LTD. (Kakegawa-shi, Japan). Polysorbate 80 (PS80), Protasorb™ O-20-NF, was purchased from Protameen Chemicals (Totowa, NJ, USA). NMR Deuterium oxide (D_2_O) 99.8% purity was purchased from Carlo Erba reagents S.A.S. (Milano, Italy). Sodium dihydrogenphosphate (NaH_2_PO_4_), disodium hydrogen phosphate (Na_2_HPO_4_), sodium chloride (NaCl), calcium chloride dihydrate (CaCl_2_·2H_2_O), magnesium chloride (MgCl_2_), potassium chloride (KCl), sodium bicarbonate (NaHCO_3_), and acetonitrile were purchased from Sigma Aldrich (Saint-Quentin Fallavier, France). Travatan^®^ commercial solution was purchased from Alcon^®^ (Hünenberg, Switzerland).

### 2.2. Fluorescence Spectroscopy

Pyrene is a lipophilic fluorescent probe often used to determine the critical micellar concentration (CMC) of surfactants. Indeed, it has the interesting feature of being sensitive to the polarity of its environment and partitioning into micelles leading to an hyperchromicity [26]. First, 40 mg of pyrene were dissolved into 10 mL of ethanol, this solution was diluted in water to achieve a stock solution with a concentration of 7.5 µM. This stock solution was used as solvent to prepare PS80 and NaHA solutions of different concentrations in triplicate. Fluorescence measurements were performed using a Varian Cary Eclipse fluorescence spectrophotometer and processed with the Cary Eclipse software version 1.1.132 (Agilent Technologies, Santa Clara, CA, USA). The data were acquired using an excitation wavelength of 230 nm with a 20 nm slit and by recording an emission scan from 350 to 450 nm with a 2.5 nm slit. Peaks 1 and 3, used to determine the fluorescence ratio I_1_/I_3_, were at approximately 372 and 383 nm, respectively. PS80 CMC was determined using a sigmoidal non-linear variable slope regression performed using GraphPad Prism version 8.0.2 for windows (GraphPad Software, San Diego, CA, USA). CMC was considered as the abscissa of the point at which the top of the sigmoid and the tangent to the inflection point crossed.

### 2.3. Differential Scanning Calorimetry (DSC)

The differential scanning calorimetry (DSC) studies were performed using a DSC 3 thermal analysis device (Mettler-Toledo GmbH, Greinfensee, Switzerland) calibrated upstream with high-purity zinc (99.99%, T_fus_ = 419.6 ± 0.7 °C, Δ_fus_H = 107.5 ± 3.2 J.g^−1^), high-purity indium (99.99%, T_fus_ = 156.6 ± 0.3 °C, Δ_fus_H = 28.45 ± 0.60 J.g^−1^) and controlled with both, and Milli-Q water (Merck Millipore, Burlington, MA, USA). Each sample was introduced in a 100 µL aluminum pan which was hermetically sealed afterwards. The conditions of the sample preparation, as well as the applied temperature program, were strictly identical to avoid any thermal history dependence of the samples. The cycle, performed twice on each sample, was freezing from ambient temperature to −80 °C at a rate of 5 °C/min, then holding for 3 min at −80 °C, then heating to 25 °C, at a rate of 5 °C/min under a 60 mL/min dry air flow. Travoprost, NaHA and PS80 were analyzed either alone or as binary drug:excipient blends at different weight ratios (1:3, 1:1 and 3:1, respectively). Measurements were performed on 3 independent samples (*n* = 3). Thermogram evaluation was performed using STARe version 16.30 (Mettler-Toledo GmbH, Greinfensee, Switzerland).

### 2.4. Nuclear Magnetic Resonance (NMR) Spectroscopy

As the pharmacological class lead, Latanoprost was used for the NMR studies to better transpose results to other prostaglandin analogues. NMR analysis solutions of PS80, latanoprost, as well as 50:50 mixtures, were prepared in D_2_O. Firstly, 1 mg of liquid raw material or mixture was weighed in a 1.5 mL Eppendorf tube. Secondly, 0.6 mL of D_2_O was added before vortexing for 2 min. Samples were left resting for 24 h at ambient temperature then vortexed again 2 min prior to transferring into NMR tubes. NMR acquisition was performed using a Bruker NMR Spectrometer (Bruker, Fällanden, Switzerland) operating at a ^1^H Larmor frequency of 600 MHz, at 25 °C, using a BBI probe. Data were treated using the software Bruker TopSpin version 4.0.9. Samples were studied by one dimensional ^1^H NMR, as well as ^1^H-^1^H correlation spectroscopy (COSY) and ^1^H-^13^C heteronuclear single-quantum coherence spectroscopy (HSQC) for signal attribution. Next, 2D rotating frame Overhauser effect spectroscopy (ROESY) experiments were performed to assess intermolecular hydrogen proximity with a mixing time of 600 ms, a recovery delay of 1.5 s, and 256 time increments with 320 scans per increment. The ^1^H NMR spectra were referenced to the resonance line of HOD at 4.70 ppm.

### 2.5. Preparation of the Eye Drops

The preservative-free eye drops (PFED) were prepared by a two-step method as described in the patent US 2014 0228364A1 [27]. First, 400 mg of sodium dihydrogen phosphate, 474 mg of disodium hydrogen phosphate and 473 mg of sodium chloride were dissolved in 100 mL of water for injection. Then, 30 mL of this buffer was used to solubilize 100 mg of NaHA, under magnetic stirring for at least 6 h. Four mg of travoprost and 100 mg of PS80 were added to the NaHA solution and mixed for 30 min until complete dissolution. The last 70 mL of the phosphate buffer were added to this travoprost solution and mixed for 90 min. Finally, pH was adjusted to 6.7 with 0.5 M NaOH or HCl and the solution was filtered through a 0.22 µm polyethersulfone (PES) membrane. Final concentrations were 40 µg/mL of travoprost, and 1 mg/mL of sodium hyaluronate and PS80, as shown in Table 1.

### 2.6. Stability Study

#### 2.6.1. Design of the Study

The PFED formulation and the comparator were stored in climatic chambers at different temperatures (2–8 °C, 25 °C and 40 °C), protected from daylight. The solutions were assayed by HPLC-UV. Travoprost content, 5,6-trans isomer, 15-keto derivative, and total impurities were used to assess the stability of the eye drops in accordance with the USP monograph for travoprost ophthalmic solutions [28].

#### 2.6.2. Travoprost HPLC-UV Assay

Samples were diluted at 1:2 in acetonitrile and filtered through a 0.22 µm PTFE syringe filter prior to analysis. Analysis was performed with an HP Agilent 1100 series HPLC system (Santa Clara, CA, USA) and a Raptor C18 5 µm 4.6 × 150 mm column purchased from Restek (Center county, PA, USA). HPLC was carried out at 40 °C with a mobile phase flow rate of 1 mL/min. The mobile phase composition was a gradient of Acetonitrile/Phosphate buffer 10 mM pH 3 from 38/62 to 50/50 *v*/*v* from 0 to 26 min, then held at 50/50 *v*/*v* from 26 to 29 min, and finally going back to 38/62 *v*/*v* from 29 to 40 min. The detection wavelength was set at 200 nm. The calibration curve ranged from 5 to 25 µg/mL. Linearity was assessed by a coefficient of determination R^2^ > 0.995, as well as a back-calculated bias of less than 5% for each concentration point of the standard curve. Accuracy was evaluated by a recovery of control samples at a target concentration between 95 and 105%. Precision was confirmed with a coefficient of variation inferior to 3% for control samples. Repeatability of injection was inferior to 0.4%.

### 2.7. Rheological Studies

Dynamic viscosity, η, was measured for the PFED formulation and Travatan^®^. In order to have a more biorelevant measure, samples were diluted in simulated tear fluid (STF), mimicking the dilution upon instillation. A Sample:STF ratio of 30:7 was used [29,30,31]. The composition of STF used is detailed in Table 2 below [32,33].

Rotational measurements were carried out on a controlled shear rate MCR102 Rheometer and data were analyzed using the Rheocompass^™^ software version 1.25 (Anton Paar, Graz, Austria). A stainless-steel cone-plate geometry (1° angle, 50 mm diameter, 100 µm gap) was used. Measurements were performed at physiological eye temperature (35 °C) for a shear rate ranging from 1 to 3000 s^−1^. Results are presented as the mean ± SD of 6 independent experiments.

### 2.8. In Vivo Studies

#### 2.8.1. Design of the Study

The ocular pharmacokinetic study was performed with 42 pigmented Dutch Belted female rabbits, aged 12 to 14 weeks and separated into two groups, one treated with Travatan^®^ and one treated with PFED. Instillation of 30 µL was completed in both eyes at t = 0 h then, at times 0.25, 0.5, 1, 1.5, 2, 3 and 4 h, three rabbits were euthanized and the aqueous humour (AH), cornea, and iris-ciliary body (ICB) were dissected and assayed for travoprost acid, the active moiety. Cornea, iris and conjunctivae of each eye were examined and rated using the Draize’s scale [34]. At each point the mean and standard deviation of 6 eyes was represented. To assess the difference between PFED and Travatan^®^_,_ an unpaired t-test with Welch’s correction was performed using GraphPad Prism version 8.0.2 for windows (GraphPad Software, San Diego, CA, USA). The area under curve (AUC) was calculated from 0.25 to 4 h using the linear-up log-down trapezoidal rule [35]. The elimination half-life (t_1/2_) for cornea and aqueous humour was determined with the log-linear regression of the tissue concentrations–time curve terminal phase. Composition of Travatan^®^ is presented in Table 3 below.

#### 2.8.2. Travoprost acid RRLC-MS/MS Assay

Travoprost acid was quantified in rabbit cornea, iris-ciliary body and aqueous humour using an RRLC-MS/MS method. Travoprost acid was extracted from the samples using ethyl acetate, the obtained solution was dried and reconstituted in 50/50 methanol/water. Chromatographic separation was performed on a Halo C18 2.7 µm 2.1 × 100 mm column (Interchim, San Diego, CA, USA) using an Agilent 1200 series rapid-resolution liquid chromatography (RR-LC) system (Santa Clara, CA, USA). A 10 µL sample was injected and eluted using a mobile phase composed of 40/60 formic acid 0.1% and a mix of acetonitrile/methanol 350/200 *v*/*v* at a 0.2 mL/min flow rate. Mass spectrometry detection was performed using an Agilent triple quadrupole 6410 (Santa Clara, CA, USA) with a capillary voltage of 4 kV, a nebulization pressure of 20 psi and a drying gas at 350 °C with a 7 mL/min flow. Calibration curves ranged from 0.05 to 10 ng/10 µL injected, with a coefficient of determination R^2^ > 0.998.

## 3. Results

### 3.1. Preformulation Studies

#### 3.1.1. Fluorescence Spectroscopy

Pyrene was chosen as a probe to evaluate the micelle formation of the pharmaceutical compounds of interest. Indeed, the fluorescence emission signal of pyrene at the 383 nm wavelength (peak 3) varied according to the polarity of its micro-environment, while the signal at 372 nm (peak 1) remained constant. Therefore, taking the ratio between the intensity at these two wavelengths gave an idea of pyrene microenvironment polarity. When pyrene was in a polar environment (e.g., water) the ratio of fluorescence between the peaks 1 and 3 (I_1_/I_3_) was higher than when in a non-polar environment (e.g., inside micelles) (Figure 1A). Accordingly, pyrene was used as a fluorescent probe in the presence of increasing amounts of excipients to evaluate micelle formation and hydrophobic interactions (Figure 1B).

On one hand, fluorescence studies showed that an increasing amount of NaHA did not display any interaction with pyrene, with the I_1_/I_3_ ratio remaining unchanged over the range of studied NaHA concentrations. On the other hand, the decrease in I_1_/I_3_ fluorescence ratio with increasing amounts of PS80 testified to micelle formation. The CMC was the concentration at which the pyrene started to be encapsulated, so the concentration at which the I_1_/I_3_ fluorescence ratio began to decrease. PS80 CMC was determined to be 0.019 mg/mL (0.015 mM) using the parameters of the sigmoidal fit with R^2^ = 0.9628. Hence, PS80 can be used as a solubilizer for water-insoluble drugs at concentrations above 0.019 mg/mL.

#### 3.1.2. Differential Scanning Calorimetry Studies

The physicochemical interactions between NaHA and travoprost, or PS80 and travoprost, were then investigated using differential scanning calorimetry (DSC). This widespread method allows the assessment of heat capacity change during the second-order phase transition. Differences in samples’ thermal behaviors when studied separately or as a mixture can provide some relevant information in regard to physicochemical interactions [36]. The thermograms obtained during the calorimetric study are presented in Figure 2.

A thermal event upon heating for the travoprost is displayed at around −22 °C (ratio 1:0). The latter event was found to be reversible, as it remained identical after applying twice the same heating setpoint range to each sample and a corresponding event occurred during the cooling process (Appendix A). The shape of the signal was characteristic of a glass transition (with an associated glass transition temperature, namely Tg, that could be determined at the inflexion point of the downward part of the curve) with relaxation (the small endothermal peak observed after the glass transition) rather than a melting peak, as there was no return to the baseline, resulting in a change in the heat capacity (ΔCp) after the transition. In addition, varying rates of cooling (20, 10 and 5 K/min) did not impact the presence of a relaxation peak (Appendix A). Mixtures of travoprost with NaHA (Figure 2A) displayed a glass transition with decreasing intensity when increasing the excipient content. This can be explained by a dilution effect as the ΔCp relative to the mass of travoprost remained constant as well as the Tg (Table 4). On the other hand, thermograms in the presence of PS80 showed a shift in the Tg, from −22 to −58 °C (Figure 2B) when increasing the excipient content. This shift and the increase in ΔCp in presence of increasing amounts of PS80 both testify to physicochemical interactions between travoprost and PS80. The latter results are confirmed by the endothermic signal of PS80 that shifted towards lower temperatures with the higher content of travoprost (Figure 2B), otherwise known as the eutectic effect [37,38].

#### 3.1.3. NMR Spectroscopy

To go deeper in the interaction study between prostaglandin analogues and PS80, NMR spectroscopy was used. First, one dimensional ^1^H and 2D COSY, HSQC, and ROESY were used to assign ^1^H signals of latanoprost and PS80 in their free forms. Then, 2D COSY and ROESY experiments were performed on latanoprost:PS80 mixtures to determine spatially close ^1^H and supramolecular association.

NMR spectra for latanoprost and PS80, as well as their molecular structures are presented in Figure 3. below. ^1^H signal attributions for latanoprost and PS80 are summarized in Table 5 below.

There are different ways for ^1^H to be close to one another and display a dipolar coupling in ROESY. First, they are close on the backbone of a single molecule. In this case a scalar coupling can be seen on the 2D COSY experiments as well. Next, if a dipolar coupling is shown on the ROESY experiments but there is no scalar coupling on the 2D COSY it can be protons from the same molecule, either intramolecular or intermolecular in a self-assembly. This phenomenon can be seen on the ROESY of samples containing PS80 or latanoprost alone. Finally, dipolar couplings seen only on the ROESY of the mixture highlight ^1^H proximity to the two kinds of molecules in the mixture.

The ROESY acquisition of a latanoprost:PS80 mixture is presented in Figure 4. In red are the dipolar couplings of the ROESY that were not observed in the COSY of the mixture nor the ROESY of latanoprost or PS80 alone. This analysis clearly shows a coupling of the aromatic cycle (L_1–2_) and the ester (L_8_) from latanoprost with the long alkyl chain (b) of PS80, as well as a coupling between the cyclopentyl (L_5_) of latanoprost with the ester (e) of PS80.

### 3.2. Formulation Studies

Preliminary formulation tests were conducted to determine optimal PS80 concentration in the final product. A 1 mg/mL concentration was kept as it was the lowest concentration enabling satisfactory solubilization of travoprost, as well as limiting its non-specific adsorption on sterile PES filters (data not show). A 1 mg/mL sodium hyaluronate concentration was used based on prior knowledge. Phosphate buffer was added to the formula to maintain pH at an eye-compatible value of 6.7 and NaCl was added to reach isotonic conditions. This formulation is further referred as PFED for the rest of the study below.

#### 3.2.1. Stability Study

As recommended by the ICH Q1A(R2), the stability of drug products must be evaluated to ensure their safe and efficient use. The stability of PFED was compared to the commercial product Travatan^®^ under accelerated (40 °C) and long-term (25 °C or 2–8 °C) conditions for 12 months, looking for a loss in travoprost content or an increase in degradation products. During the entirety of the stability studies, the amount of 15-keto derivative remained below the limit of detection for both PFED and the comparator. In contrast, the 5,6-trans isomer and the total impurities were quantifiable but persisted within the acceptance criteria of the USP monograph for travoprost ophthalmic solutions (Figure 5A,B). The storage of PFED formulation at 40 °C showed a decrease in content after 3 months (Figure 5C) while Travatan^®^ remained stable for up to 12 months. However, PFED remained stable for 12 months at an ambient temperature, meeting the claims of the comparator, Travatan^®^.

#### 3.2.2. Rheological Studies

The rheological behavior of both PFED and Travatan^®^, after the simulation of instillation dilution, was studied. The flow curves are presented in Figure 6 for both the PFED formula and Travatan^®^ after a dilution in STF at 35 °C. This flow curve gives an idea of both resting rheological properties and properties under shear stimulation. For Travatan^®^, the viscosity was low and remained quite constant over the shear rate range, at 0.9 mPa·s. This result is close to the dynamic viscosity of water and has the same Newtonian behavior. On the other hand, PFED had a higher viscosity, starting with a Newtonian plateau at 5.1 mPa·s and then displaying a shear-thinning behavior starting at approximately 100 s^−1^, reaching a final viscosity of 3.1 mPa·s at a shear rate of 3000 s^−1^. In other words, after instillation, PFED had a 5-fold higher dynamic viscosity than Travatan^®^ when resting, which makes it less prone to draining into the lacrimal duct.

#### 3.2.3. Pharmacokinetic Study

A pharmacokinetic study was performed on rabbits to evaluate the ocular distribution of travoprost acid (active moiety of travoprost) after the single instillation of PFED or Travatan^®^. Animal body weights recorded at baseline and day of euthanasia were within a normal range, no abnormal behavior or unhealthy signs were reported during the in-life part of the study. Ocular tolerance evaluated by the Draize’s scale were similar at baseline and after both treatments for cornea and conjunctiva.

After single instillation, the amount of travoprost acid in the cornea reached a maximum for the first timepoint studied (T_max_ = 0.25 h), with a Cmax of 568 and 1331 ng/g for the comparator Travatan^®^ and PFED (Table 6). From there, the kinetic decreased with comparable elimination t_1/2_ of 1.04 and 1.05 h, respectively, consequently showing that the difference observed in AUC_0.25–4 h_ was primarily due to a difference in the early absorption rate of both products (Figure 7A). Consequently, the amount of travoprost acid in the aqueous humour was greater for PFED than for Travatan^®^ with, for each, AUC_0.25–4 h_ of 151 and 68 ng/mL·h as it was related to the uptake of travoprost acid in the cornea. Similarly, the amount of travoprost acid reaching the iris-ciliary body increased 3.2 times with PFED compared to the Travatan^®^, with AUC_0.25–4 h_ of 68 and 21 ng/g·h, respectively (Table 6).

## 4. Discussion

We presented, in the first part of this article, preformulation studies regarding the key excipients of the formula NaHA and PS80, and the active substance. To begin, we chose pyrene as a fluorescent probe because its LogP is 6.0 [39]. Indeed, it was interesting to mimic the behavior of lipophilic drugs such as prostaglandin analogues. This first study showed that the excipient NaHA did not exhibit any affinity for it, therefore showing its lack of physicochemical interaction with liposoluble drugs. This finding was further confirmed by the DSC studies. Indeed, no shifts in the Tg of travoprost occurred with varying ratios of NaHA:Travoprost. This method has been successfully used to predict the compatibility of components within pharmaceutical mixtures [40,41,42]. As no incompatibility was brought to light between travoprost and NaHA, the latter was further used in the final PFED formulation.

On the other hand, for PS80, the fluorescence studies showed a decrease in the I_1_/I_3_ ratio with increasing PS80 concentrations. The determined CMC of PS80 was approximately 0.019 mg/mL (or 0.015 mM), which is consistent with the literature, ranging from 0.018 mM, determined by surface tension measurement [43], to 0.015 mM obtained with a fluorescence-based method using the derivative pyrene-3-carboxaldehyde [44]. Therefore, the observed effect can be attributed to PS80 micelles formation, into which pyrene is integrated, showing PS80 potential to solubilize lipophilic drugs given concentrations higher than the CMC.

Furthermore, DSC studies showed a clear effect of the presence of PS80 on the thermal events. The shift in Tg towards lower values of temperature was similar to the effect of plasticizers in a polymer melt, which testifies to a miscibility of the different molecules as well as an improvement in solubility [45,46]. This demonstrated a physicochemical interaction between PS80 and prostaglandin analogues. This result is in accordance with the eutectic effect that was highlighted in the Results section, related to PS80 thermal behavior when mixed with travoprost. PS80 is a non-ionic surfactant, meaning it has both a lipophilic and a hydrophilic moiety. When its concentration is high enough it self-assembles into micelles, grouping the lipophilic parts in its core and exposing the hydrophilic fragment to the surrounding water [47].

This was further clarified by the NMR studies. Indeed, NMR has previously been used to assess the molecular organization between prostaglandin analogues and solubilizers such as cyclodextrins [48] or other surfactants [10]. The orientation suggested in this work seems to be similar to the one explicated by Ochiai et al., suggesting that in micelles of BAK and PEG-Stearate the latanoprost is encapsulated in the hydrophobic core with a preferential orientation, presenting the cyclopentyl fragment towards the hydrophilic parts of the surfactants, therefore limiting its exposure to water [10]. Besides, NMR studies conducted with NaHA and latanoprost did not display physicochemical interactions in solution.

In the second part of this work, we studied a final product: PFED containing travoprost as an active substance, solubilized and stabilized by PS80, with NaHA as a thickener and potential cytoprotective agent. PS80 has already been used in eye drops as a solubilizer for cyclosporine [18], and this work confirmed its potential to further solubilize prostaglandin analogues in drug delivery while maintaining stability in solution.

According to the ICH guideline Q1A(R2) on stability studies for drug products, a period of 12 months is the minimum that has to be covered at room temperature (25 °C) at the time of submission [49]. As the stability results at 40 °C at 12 months were not within the USP criteria, the stability studies at room temperature were the bare minimum for a submission, but would have to be continued for longer periods of time to secure long-term stability data. So far, however, the PS80 and NaHA seem to be satisfying excipients regarding the travoprost stability at room temperature.

Rotational rheological studies were performed to assess the dynamic viscosity of PFED and Travatan^®^ upon instillation. Travatan^®^ exhibited a Newtonian-like behavior with a constant viscosity over the shear rate range analyzed. The mean viscosity of 0.9 mPa·s at 35 °C was slightly higher than pure water at this temperature (0.7 mPa·s) [50]. Therefore, it is expected that, upon instillation, Travatan^®^ will rapidly mix with tear fluid [51,52,53]. The first challenge in ocular drug delivery is to overcome the high clearance of lacrimal fluid upon instillation. It has been reported that the tear fluid is entirely renewed within 2–3 min under physiological conditions, therefore leading to a fast clearance of the eye drops [54,55]. On the other hand, an increase in viscosity, as displayed for PFED, can lengthen the residence time of the eye drop on the ocular surface [30,55,56,57]. Indeed, NaHA was added to the eye drop formulation to thicken the solution. As expected, adding a high molecular weight polymer increased the viscosity at low shear and turned the solution into a shear-thinning fluid [58,59].

The in vivo studies conducted on rabbits exhibited an increase in AUC for the three studied structures of the eye, cornea, AH, and ICB. AUC represents the total drug exposure over the studied period. In our study, this improvement in AUC seemed to be linked to a greater absorption of travoprost, as the elimination constants were similar for the PFED and the comparator. This greater AUC observed after single-drop instillation of PFED related to Travatan^®^ (between 2.2- and 3.2-fold for the AH and the ICB, respectively) could be linked to its rheological behavior. As it was discussed for dry eye formulations containing 0.1% NaHA [60], the rheological properties of the final dosage can improve its in vivo residence time, therefore extending the exposure time of the cornea to the drug product [30,52,53]. Another mechanism could be involved into this enhanced ocular bioavailability. Indeed, PS80 has been reported to improve travoprost uptake in vivo in rabbits in a nano-emulsion drug formulation compared to Travatan^®^ [61]. Further studies, such as a permeability study or an evaluation of the residence time, could unveil the relative contribution of these two mechanisms on the AUC increase. With such an increase in ocular bioavailability, the therapeutic dose of travoprost could potentially be lowered [62,63]; therefore, limiting systemic exposition and side effects while maintaining the therapeutic effect.

## 5. Conclusions

In this work, PS80 and sodium hyaluronate were assessed as excipients in a preservative-free formulation of travoprost. PS80 was demonstrated to be a successful option to solubilize and stabilize travoprost in aqueous solution, while NaHA did not display any physicochemical interactions with travoprost and enhanced the rheological behavior of the formula. Therefore, we showed that NaHA is a good candidate as a thickener and a cytoprotective agent. The final product was stable for at least a year at room temperature and outperformed the comparator in terms of ocular bioavailability. These encouraging results demonstrate the potential of such a drug delivery system. However, further studies should be performed in order to evaluate the origin of the increase in ocular bioavailability, and identify whether it is caused by an extended residence time on the ocular surface, or a permeation enhancement caused by PS80. Ultimately, trying to lower the dose to reach the same therapeutic effect or ocular bioavailability as the comparator while reducing the risk of side effects could be greatly beneficial to patients, and improve their adherence as well as disease control.

## Figures and Tables

**Figure 1 pharmaceutics-14-00453-f001:**
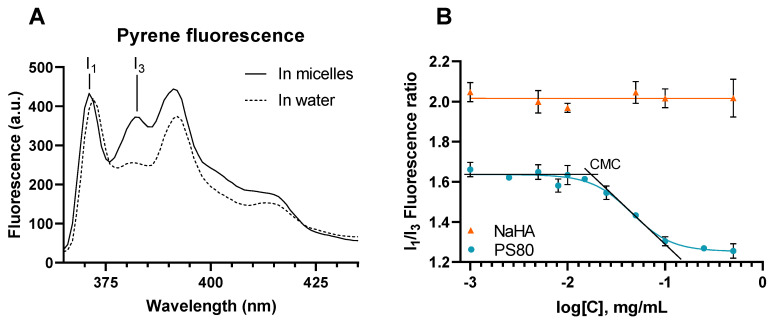
(**A**) Emission scan of pyrene in absence of surfactants (in water) or in presence of surfactants above the CMC (in micelles) showing the change in I_1_ and I_3_ ratio. (**B**) I_1_/I_3_ fluorescence ratio as a function of the concentration NaHA (▲) and PS80 (●) with the graphical illustration for the determination of the CMC.

**Figure 2 pharmaceutics-14-00453-f002:**
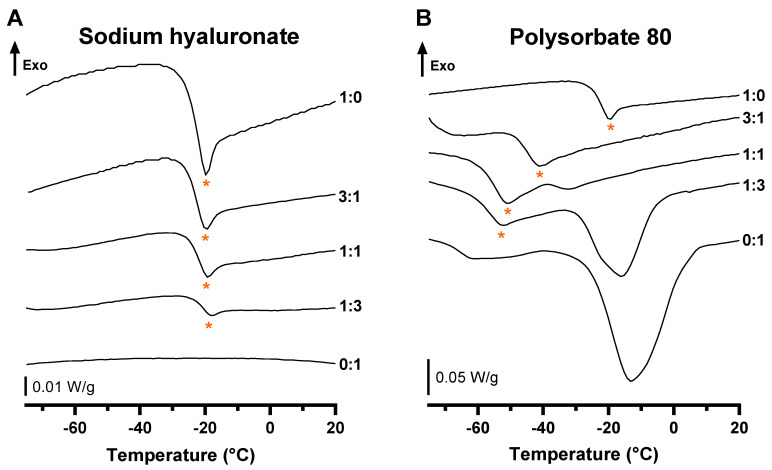
DSC Heating curves of Travoprost:Excipient mixtures with ratios from top to bottom: 1:0, 3:1, 1:1, 1:3 and 0:1 for (**A**) NaHA and (**B**) PS80, the glass transition with relaxation of travoprost is marked by a star. Displayed thermograms were shifted for clarity.

**Figure 3 pharmaceutics-14-00453-f003:**
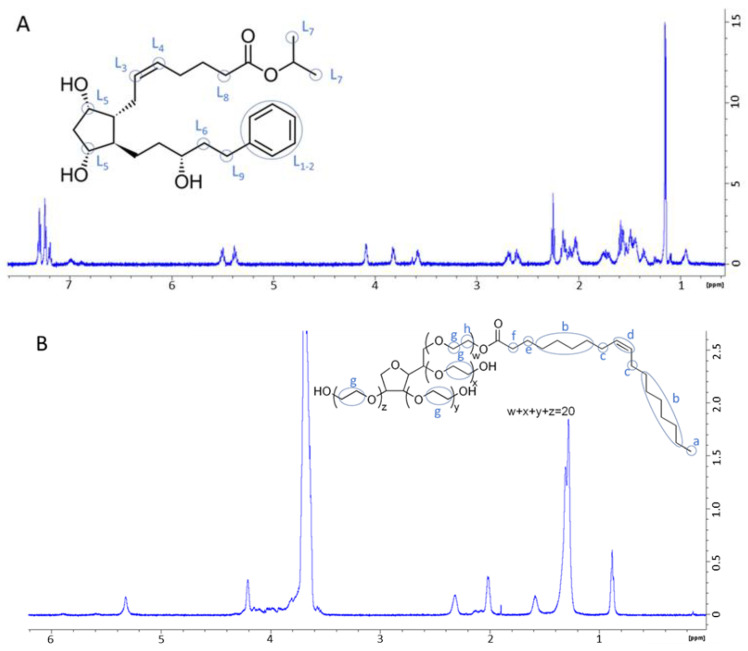
Molecular formula and ^1^H-NMR spectra of (**A**) Latanoprost, with ^1^H groups from L_1_ to L_9_ according to Table 5 and (**B**) Polysorbate 80, with ^1^H groups from a to g according to Table 5, in D_2_O.

**Figure 4 pharmaceutics-14-00453-f004:**
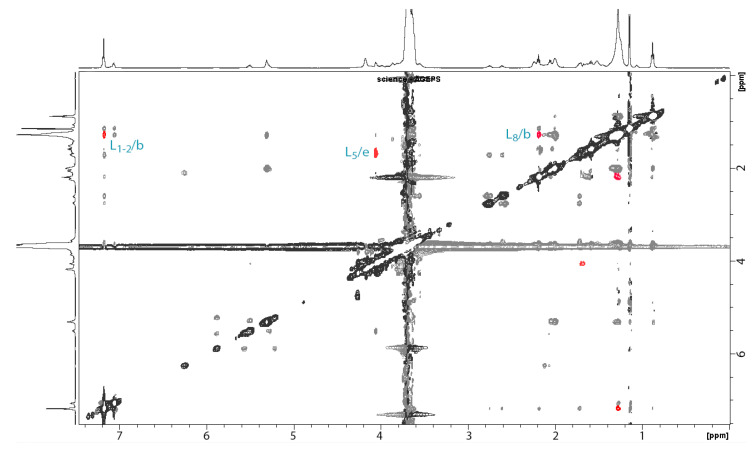
ROESY experiment of the latanoprost:PS80 mixtures, highlighted in red are the dipolar couplings not shown in the ROESY of either the latanoprost or PS80 alone.

**Figure 5 pharmaceutics-14-00453-f005:**
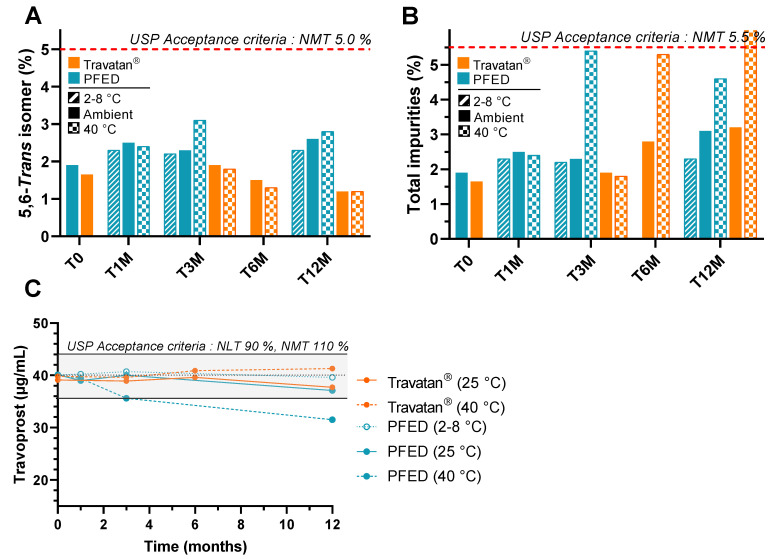
Dosage over time of (**A**) 5,6-Trans isomer, (**B**) total impurities, (**C**) travoprost, in both the comparator, Travatan^®^, (in orange) at 25 °C and 40 °C and the PFED (in blue) at 2–8 °C, 25 °C and 40 °C. NLT: not less than; NMT: not more than; PFED: preservative-free eye drops; USP: United States Pharmacopoeia.

**Figure 6 pharmaceutics-14-00453-f006:**
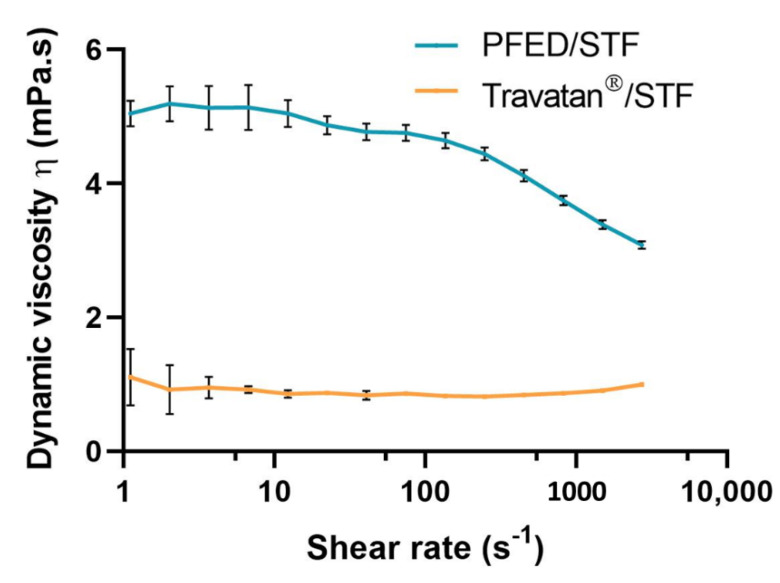
Flow curves of preservative-free eye drops (PFED) and Travatan^®^ after dilution in simulated tear fluid (STF) at a 30:7 ratio, results are presented as mean ± SD of 6 independent experiments.

**Figure 7 pharmaceutics-14-00453-f007:**
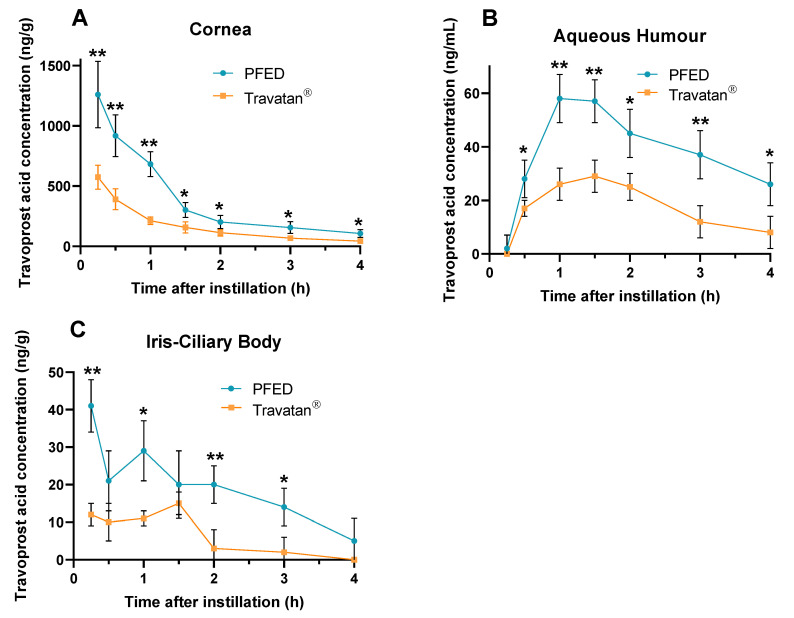
RRLC-MS Dosage of travoprost acid after a single instillation to rabbits, presented as mean ± SD (*n* = 6) over time. (**A**) in cornea, (**B**) in aqueous humour, (**C**) in iris-ciliary body. PFED: preservative-free eye drops; * *p*-value < 0.01%, ** *p*-value < 0.001%.

**Table 1 pharmaceutics-14-00453-t001:** Final composition of the preservative-free eye drops, NaHA: Sodium Hyaluronate, PS80: Polysorbate 80.

Component	Composition for 100 mL	Composition in % *w*/*v*
Travoprost	4 mg	0.004
NaHA	100 mg	0.1
PS80	100 mg	0.1
NaH_2_PO_4_	400 mg	0.4
Na_2_HPO_4_	474 mg	0.474
Sodium Chloride	473 mg	0.473
Water for injection	QSP 100 mL	QSP 100

**Table 2 pharmaceutics-14-00453-t002:** Composition of simulated tear fluid (STF) used to mimic lacrimal dilution upon instillation.

Component	Composition for 100 mL
CaCl_2_.2H_2_O	8 mg
MgCl_2_	5 mg
KCl	138 mg
NaCl	670 mg
NaHCO_3_	200 mg
Water for injection	QSP 100 mL

**Table 3 pharmaceutics-14-00453-t003:** Composition of commercial product Travatan^®^, ND: Non-disclosed.

Component	Composition in % *w*/*v*
*Active Pharmaceutical Ingredient*	
Travoprost	0.004
*Excipients with known effect*	
Polyquaternium-1	0.001
Propylene glycol	0.00075
Polyoxyethylene hydrogenated castor oil 40 (HCO-40)	0.0002
*Other excipients*	
Boric acid, Mannitol, NaCl	ND
Purified Water	QSP 100 mL

**Table 4 pharmaceutics-14-00453-t004:** Parameters after “parallel tangents method” (examples of this method presented Appendix A) applied to the events of interest for mixtures of Travoprost:Excipient. Tg: Glass Transition temperature, ΔCp: Change in heat capacity normalized to travoprost content, *n* = 3.

Mass Ratio	Sodium Hyaluronate		Polysorbate 80	
	Tg (°C)	ΔCp (J/g/K)	Tg (°C)	ΔCp (J/g/K)
1:0	−21.9 ± 0.6	−0.57 ± 0.05	−21.9 ± 0.6	−0.57 ± 0.05
3:1	−20.3 ± 0.6	−0.58 ± 0.06	−44.2 ± 0.7	−1.32 ± 0.06
1:1	−21.4 ± 0.6	−0.60 ± 0.05	−54.4 ± 0.6	−2.04 ± 0.06
1:3	−22.3 ± 0.7	−0.55 ± 0.05	−58.8 ± 0.7	−2.96 ± 0.07

**Table 5 pharmaceutics-14-00453-t005:** ^1^H NMR signal attribution for PS80 and Latanoprost in D_2_O in regard of the molecular structure shown in Figure 3.

Polysorbate 80									
δ (ppm)		5.32	4.20	3.69	2.31	2.02	1.58	1.28	0.88
Attribution		d	h	g	f	c	e	b	a
**Latanoprost**									
δ (ppm)	7.36–7.26	5.56	5.45	4.16	3.88	2.67	2.32	1.56	1.22
Attribution	L_1–2_	L_3_	L_4_	L_5_	L_5_	L_9_	L_8_	L_6_	L_7_

**Table 6 pharmaceutics-14-00453-t006:** Pharmacokinetic parameters observed after single instillation. AH: aqueous humour; AUC: area under curve; Cmax: maximal concentration reached; ICB: iris-ciliary body; PFED: preservative-free eye drops; t_1/2_: elimination half-life; Tmax: time for maximal concentration; N/A: calculation of t_1/2_ was not relevant for ICB data. Results are presented as mean ± SD.

Pharmacokinetic Parameters	Travatan^®^			PFED		
	Cornea	AH	ICB	Cornea	AH	ICB
AUC_0.25–4 h_ (ng/g·h)	568 ± 40	68 ± 7	21 ± 4	1331 ± 88	151 ± 10	68 ± 7
Tmax (h)	0.25	1.5	1.5	0.25	1.0	0.25
Cmax (ng/g)	574 ± 99	29 ± 6	15 ± 3	1260 ± 275	58 ± 9	41 ± 7
t_1/2_ (h)	1.04 ± 0.1	1.27 ± 0.3	N/A	1.05 ± 0.2	2.34 ± 0.5	N/A

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
