# Peer review of "New Preservative-Free Formulation for the Enhanced Ocular Bioavailability of Prostaglandin Analogues in Glaucoma"

_pharmaceutics, 2022, doi:10.3390/pharmaceutics14020453_

Round 1

Reviewer 1 Report

The submitted manuscript advances most promising formulations for travoprost that avoid secondary effects while increasing the drug bioavailability. A bunch of complementary experimental techniques are utilized as arguments. The pharmaceutical side of the manuscript is convincing. However, its basic research side needs improvement: 

  • No explanation is offered for the use of latanoprost in NMR experiments, while all other investigations / formulations involve travoprost. Important information concerning NMR studies is missing.
  • DSC studies need important clarifications concerning experimental details, data processing and significance of the obtained results.

The attached reviewed version of the manuscript contains several highlighted zones with detailed comments and suggestions. (There are also a few minor editing errors that have to be addressed.) The authors may chose to simply eliminate the problematic sections or re-edit them, possibly with the help of additional experimental data. Either way, their effort would result in a major revision of the present manuscript. In its submitted form this is not publishable in Pharmaceutics.

Author Response

Dear reviewer,

Thank you for your valuable comments and suggestions on our manuscript. We believe the article is of better quality now thanks to you. Please find below the answers to some of the comments made in the document. Detail of changes are tracked in the re-submitted manuscript.

Best regards,

  • DSC studies need important clarifications concerning experimental details, data processing and significance of the obtained results.

We further clarified the methods for DSC studies (Added cooling rate, time spent at -80 °C, number of cycles performed per sample and number of independent samples). Moreover, we added supplementary experimental data to back up statements of the article and avoid such formulations as “data not shown”. Supplementary Figure 1 shows reversibility of thermal event (with 2 cycles on 1 sample, and cooling thermograms), supplementary Figure 2 shows the persistence of relaxation peak regarding three different cooling rates, and supplementary Figure 3 illustrates the integration of the thermal events using the parallel tangents method.
Furthermore, we edited Figure 2 as you suggested by adding the missing y-scale on the bottom left, extending the x-axis range to better visualize the baselines on the left-hand side, and mentioned that the thermograms were shifted for better clarity.

  • In the absence of any specific statement, one has to infer that DSC studies pertain to pure (liquid) components and their blends. If this is true, DSC systems are quite different, thus their associated data are of severely limited use for the actual pharmaceutical formulations!
  • What was NOT demonstrated by DSC is the PS80 - drug interaction in conditions as close as possible to the pharmaceutical formulations. This was proven in many systems involving micelles formation and drug capture. A true "calorimetric study", involving PS80 among other excipients, was done by Sarpietro et al, Mol. Pharmaceutics 2011, 8, 642–650

Indeed, the systems studied by DSC are different in nature from the final eyedrop formulation. We agree that the reference cited, Sapietro et al. is more relevant regarding the studied system and the final formulation. Hence, we first tried to study drug:excipient association in solution but given the low concentration in water (0.004% for travoprost) the attempt was unsuccessful. Therefore, we decided to study binary mixtures between drug and excipients as this method is commonly used to assess the physicochemical compatibility between raw materials.  In our case, the travoprost:PS80 ratio in the final pharmaceutical formulation is 1:25, i.e. between 0:1 and 1:3. Since we observed an impact on the highest PS80 content sample, one can extrapolate the same effect, at least for the travoprost:PS80 ratio in the final pharmaceutical formulation.
However, the reference from Sarpietro et al.has been introduced in the revised version of the manuscript L246.

  • The systems investigated in ref. [43,44] are quite different from the present manuscript ones (although one of the authors of this manuscript and the Mettler Toledo brand are common with ref. [43]).

The former references 43 and 44 (36 and 37 actual references) were mentioned to introduce the eutectic effect that is a generic event independent of the system studied. Consequently, these references have been moved to the results part where the eutectic effect is mentioned for the first time in the manuscript.

  • However, the proof of pyrene being captured by PS80 micelles does NOT automatically extend to travoprost

Indeed, pyrene is a probe commonly used to determine surfactants critical micellar concentration. However, this CMC is intrinsic to the surfactant itself and does not imply travoprost capture by micelles but gives roughly an idea of the lowest concentration at which there is a micellization and therefore an encapsulation phenomenon can be expected towards lipophilic drugs such as prostaglandin analogues.

  • No explanation is offered for the use of latanoprost in NMR experiments, while all other investigations / formulations involve travoprost. Important information concerning NMR studies is missing.
    Some explanation concerning the use of latanoprost for NMR (while all other experimental data refer to travoprost) is mandatory. See below further notes on NMR data.

As suggested, a more detailed description of sample preparation in the materials & methods part of NMR studies was added.
As the pharmacological class lead, latanoprost was used for the NMR studies to better transpose results to other prostaglandin analogues. Furthermore, this allowed us to compare obtained results with previously published NMR results regarding latanoprost incorporation into other surfactant micelles (Ochiai et al. Int. J. Pharm. 410:23-30).

Reviewer 2 Report

The work by Alviset et al. reports on the design of a new formulation for the local delivery of prostaglandin analogues in the treatment of glaucoma. Specifically, the authors first investigated therapeutic agent interactions with excipients and the stability of the resultant formulation and then, they studied the pharmacokinetics in an in vivo model. The work is well presented and I recommand its acceptance after minor revisions.

  1. In the introduction section, the authors state that PS80 could be better tolerated than BAK and ricin oil (line 68-69). Which is the reason?Please, add explanations to better support this statement
  2. Why was the NMR characterization performed on latanoprost instead of travoprost like all the other characterizations?Please, explain and/or justify
  3. In the preparation of the eye drops, why do the authors add the remaining volume in a second step (line 139-140)? is there a specific reason? in case, please clarify.
  4. Have you ever tested the cytocompatibility of the optimized formulation?This is a crucial point as surfactants generally induce cell apoptosys by interacting with cell membrane. For this reason, cytocompatibility tests and/or a dose-response curve should be performed and added to the manuscript. This could guide the identification of the maximum amount of PS80 that can be used and consequently, the corresponding amount of therapeutic agent
  5. Other minor remarks:

line 23: please, define the acronym PS80 at its first mention;

line 88: replace "sodim" with "sodium";

line 89: replace CaCl2.2H2O with CaCl2•2H2O;

line 154: replace 1/2 with 1:2;

line 325, 327, 329, 414: replace mPa.s with mPa•s;

line 349, 351: replace ng/ml.h with ng/ml•h;

Figure 3: please, magnify the numbers to make them more easily readable

Author Response

Dear reviewer,

We deeply appreciate the time spent commenting this manuscript and the suggestions you made to improve it. We believe we addressed all your concerns in this re-submitted version of the manuscript. Please find below the details of modifications that were made following your valuable advice.

Thanks again for the work you did reviewing this article.

Best regards,

  • In the introduction section, the authors state that PS80 could be better tolerated than BAK and ricin oil (line 68-69). Which is the reason? Please, add explanations to better support this statement

This statement was made based on another team’s previously published results, added “as brought to light by Fukuda et al. during in vitro/in vivo studies » to clarify the statement, with adequate reference citation.

  • Why was the NMR characterization performed on latanoprost instead of travoprost like all the other characterizations? Please, explain and/or justify

As the pharmacological class lead, latanoprost was used for the NMR studies to better transpose results to other prostaglandin analogues. Furthermore, this allowed us to compare obtained results with previously published NMR results regarding latanoprost incorporation into other surfactant micelles (Ochiai et al. Int. J. Pharm. 410:23-30).

  • In the preparation of the eye drops, why do the authors add the remaining volume in a second step (line 139-140)? is there a specific reason? in case, please clarify.

The mixing process was based on a literature patent (US 2014 0228364A1) operating this way with sodium hyaluronate, we prepared it the same way without trying the “all at once” approach.

  • Have you ever tested the cytocompatibility of the optimized formulation? This is a crucial point as surfactants generally induce cell apoptosis by interacting with cell membrane. For this reason, cytocompatibility tests and/or a dose-response curve should be performed and added to the manuscript. This could guide the identification of the maximum amount of PS80 that can be used and consequently, the corresponding amount of therapeutic agent

The in vitro cytotoxicity was not evaluated during this work. We agree with your statement that it could have been a logical step before going in vivo in rabbits. Yet, considering data from literature concerning polysorbate 80 cytotoxicity (Fukuda et al. 2013, Clin. Ophthalmol. 7:515-520 and Younis et al. 2008 J. Ocul. Pharmacol. Ther. 24:206-216, and Arechabal et al. 1999 J. Appl. Toxicol. 19:163-165) and knowing that we were going to perform a pharmacokinetic study in rabbits, we decided to evaluate the toxicity in vivo using the Draize test instead of doing in vitro testing on cell cultures.

  • Other minor remarks:

line 23: please, define the acronym PS80 at its first mention;
The acronym has been introduced in the related line.
line 88: replace "sodim" with "sodium";
The typo has been corrected
line 89: replace CaCl2.2H2O with CaCl2•2H2O; line 325, 327, 329, 414: replace mPa.s with mPa•s; line 349, 351: replace ng/ml.h with ng/ml•h;
The modifications have been made
line 154: replace 1/2 with 1:2;
The modification has been made.
Figure 3: please, magnify the numbers to make them more easily readable.
The numbers along the axis on both figure 3 and 4 have been magnified for better readability.

Reviewer 3 Report

The present paper purposes the preparation of preservative-free eye drops based on polysorbate 80, combined to sodium hyaluronate as viscosizing and cytoprotective agent for ocular application.

The topic is interesting and the paper is well written. Some minor revisions should be made before publication.

Lines 136-137. Please specify the amount of salt and the volume of water for injection used. If not, indicate Table 1 as reference.

Line 139: The rest…..is generic. Specify the volume

Line 141: pH was adjusted to 6.7 with….

Line 244: Sodium hyaluronate and Polysorbate 80. Use the abbreviated forms (check the manuscript).

References:

The journal names must be abbreviated. In some references this rule has not been respected.

Reference 22 is too old. Please replace it.

Author Response

Dear reviewer,

Thank you for the time you took to give us feedback on our manuscript. In the re-submitted version you will find our corrections addressing your remarks. Details of changes made are listed below, in black was your review report, and in blue our modifications.

Thanks again for your work.

Best regards,

  • Lines 136-137. Please specify the amount of salt and the volume of water for injection used. If not, indicate Table 1 as reference.
    The quantities of salt/volume have been introduced in the revised version of the manuscript.
  • Line 139: The rest….. is generic. Specify the volume
    The amount of phosphate buffer has been introduced in the revised version of the manuscript.
  • Line 141: pH was adjusted to 6.7 with….
    The information has been given in the revised version of the manuscript
  • Line 244: Sodium hyaluronate and Polysorbate 80. Use the abbreviated forms (check the manuscript).
    The abbreviations for NaHA and PS80 have been used in the corresponding paragraph.
  • References: The journal names must be abbreviated. In some references this rule has not been respected. Reference 22 is too old. Please replace it.
    The abbreviations in Zotero have been updated for adequate reference citations. Moreover, reference 22 has been removed, remaining former references 21, 23 and 24.

Round 2

Reviewer 1 Report

The authors have properly addressed all previously signaled issues. In the present form the manuscript is publishable. There may be a minor modification (parallel tangents in quotation marks, "...") that has been explained in the attached, reviewed form of the manuscript.

Author Response

Dear reviewer,

Thank you for your feedback. The quotation marks were added in the revised versions of the manuscript and supplementary data for table 3 and figure S3 as requested.

Best regards,